# A recurrent network for segmenting the thrombus on brain MRI in patients with hyper-acute ischemic stroke

**Sofia Vargas-Ibarra**[1]             SOFIA.VARGASIBARRA@UNIV-EVRY.FR
**Vincent Vigneron**[1]              VINCENT.VIGNERON@UNIV-EVRY.FR
**Hichem Maaref**[1]               HICHEM.MAAREF@UNIV-EVRY.FR
**Jonathan Kobold**[1]              JONATHAN.KOBOLD@GMX.DE
[1] *Univ Evry, Université Paris-Saclay, IBISC, Evry, France.*

**Sonia Garcia-Salicetti**[2]        SONIA.GARCIA@TELECOM-SUDPARIS.EU
[2] *SAMOVAR, Télécom SudParis, Institut Polytechnique de Paris, Palaiseau, France.*

**Nicolas Chausson**[3]             NICOLAS.CHAUSSON@CHSF.FR
**Didier Smadja**[3]               DIDIER.SMADJA@CHSF.FR
**Yann Lhermitte**[3]              YANN.LHERMITTE@CHSF.FR
[3] *Neurology Department, Sud-Francilien Hospital, Corbeil-Essonnes, France.*

**Editors:** Accepted for publication at MIDL 2024

## Abstract

In the stroke workflow, timely decision-making is crucial. Identifying, localizing, and measuring occlusive arterial thrombi during initial imaging is a critical step that triggers the choice of therapeutic treatment for optimizing vascular re-canalization. We present a recurrent model that segments the thrombus in patients suffering from a hyper-acute stroke. A cross-attention module is defined to merge the diffusion and susceptibility-weighted modalities available in magnetic resonance imaging (MRI), which are fed to a modified version of convolutional long-short-term memory (CLSTM). It detects almost all the thrombi with a Dice higher than 0.6. The lesion segmentation prediction reduces the false positives to almost zero and the performance is comparable between distal and proximal occlusions.

**Keywords:** hyperacute stroke, brain imagery, MRI modalities, lesion, thrombus, spatial recurrence, deep learning

## 1. Introduction

Stroke causes millions of deaths and cerebral palsy per day nowadays. Acute ischemic stroke (AIS) occurs when a blood clot (thrombus) blocks an artery, restricting average blood circulation and initiating brain tissue damage. The lesion caused by the stroke can be separated into (1) the penumbra, which represents the brain tissue that is potentially recoverable but temporarily injured, and (2) the core, the irreversibly damaged tissue. The lesion's size and location are crucial pieces of information that help the doctors locate the thrombi and decide the treatment. In an emergency case, the pathology is in the hyper-acute phase (<6h from the onset of the symptoms) which is critical for saving lives and improving life expectancy of patients.

This paper proposes an experimentally validated and computationally affordable method capable of segmenting the thrombus of an AIS patient with few false positives using lesion segmentation as support. This technique is based on a light spatial recurrent neural network architecture.

## 1.1. Related works on classical image processing techniques

As the lesion is brighter (in the case of DWI) or darker (in ADC) depending on the MRI modalities, some approaches exploit these properties using non-learning algorithms. For instance, unsupervised clustering such as Fuzzy C-means produces considerably good automatic segmentation results (Tsai et al., 2014). It is applied on intensity values lower than the histogram peak (healthy brain tissue) using a fixed threshold, and it eliminates the artifacts thanks to several modalities. Also, anomaly detection can be done on DWI as in (Nazari-Farsani et al., 2020), where the Crawford-Howell t-test and the comparison of stroke images to healthy controls are used to develop a classifier to discriminate the images into stroke or non-stroke categories following the lesion segmentation. For the thrombi, (Santos et al., 2014) defined an automatic segmentation method in three steps: segmentation of the contralateral vasculature; creation of a mask in the occluded arterial segment by mapping the contralateral segmentation using mirror symmetry; and segmentation of the thrombus using intensity-based region growing. In (Löber et al., 2017) Random Forests are used to classify features extracted by thresholding and connected component clustering.

## 1.2. Deep learning works related to brain imagery

ISLES challenges (Petzsche et al., 2022) provide several public datasets that allow the training of lesion segmentation models. To manage 3D inputs, 2D CNN models have been used (Clerigues et al., 2019), (Shah et al., 2020), the mixture of 2D and 3D architectures have been proposed as D-Unet (Zhou et al., 2021), DFENet (Basak et al., 2021), and also using variations of Unet3D (Omarov et al., 2022), (Ashtari et al., 2023). Attention methods such as AABTS-Net (Tian et al., 2023) have been used, incorporating axial attention (Wang et al., 2020), skip-connections as PerfUnet (de Vries et al., 2023) or soft-attention mechanisms as DRANet (Liu et al., 2020). UNETR proposed by (Hatamizadeh et al., 2021) is an architecture adapted from Visual Transformers (Dosovitskiy et al., 2021) to 3D inputs for medical applications. But, all these methods rely on considerable big datasets and neither of them is only focused on the hyperacute phase, where the delineation of the affected area is more difficult to do (Jones and Cercignani, 2010). On the other hand, for the thrombi segmentation, (Lucas et al., 2019) learns to segment potential candidates using Unet3D and a Convolutional architecture classifies the candidates. Polar-UNet (Zoetmulder et al., 2022), uses a CNN to segment thrombi restricting the volume-of-interest to the brainstem, or in (Tolhuisen et al., 2020) patches from two hemispheres are used to segment the thrombi, as a healthy brain tissue should be symmetric. However, all these methods use CT images. For MRis in (Kobold et al., 2019) Multidirectional Unet was used and a recurrent method also denoted as Logic LLSTM (Kobold, 2019) but no distal thrombi were present.

## 2. Materials and methods

## 2.1. Dataset and preprocessing

Two datasets are used in this study: CHSF (proximal occlusions (P)) and MATAR (distal occlusions (D)). All MRIs come from patients treated for stroke at the Centre Hospitalier Sud-Francilien and were acquired from a 1.5 T and a 3 T GE Healthcare MRI machine in the hyper-acute phase, before treatment. The details of datasets are described in Table 1.

**Table 1:** Dataset description. The sizes are in pixels (pix.) and the average (avg) of the ground-truths sizes are provided

| | CHSF | MATAR |
|---|---|---|
| Age (years) | $74.33 \pm 0.74$ | $72.95 \pm 1.6$ |
| Male | 43.3% | 46.5% |
| Hypertension | 62% | 59% |
| Current smokers | 11% | 13% |
| DWI-ASPECTS[1] | 3.69 (0-10) | 9 (7.75-10) |
| SWAN slices | 72 to 216 | 72 to 232 |
| DWI slices | 24 to 38 | 24 to 40 |
| SWAN shape (pix. per slice) | $512\times512$ | $512\times512$ |
| DWI shape (pix. per slice) | $256\times256$ | $256\times256$ |
| Thrombi size (mm$^3$) | 231.97 | 77.04 |
| Lesion size (mm$^3$) | 31770.21 | 5010.15 |
| Number of patients | 63 | 125 |

Even though they have been used for several clinical studies (Marnat et al., 2023), we use only a subset of it, segmented by neurologists. Notice the differences between MATAR and CHSF: the thrombi and lesion size and the ASPECT. Due to registration problems such as missing modalities, only 188 MRIs are used finally. The susceptibility-weighted angiography (SWAN) with its associated PHASE are used to segment the thrombi, and the apparent diffusion coefficient (ADC), and diffusion-weighted imaging (DWI) modalities are used to segment the lesion. The susceptibility images (SWAN and PHASE) have the same geometries and alignment. This is because SWAN is calculated using the machine's PHASE and the magnitude image. It happens the same for the diffusion ones, as ADC is obtained from DWI and B0.

The dataset is normalized using Nyul's method (Nyul et al., 2000). Computing per modality an average histogram using the quantiles of all the data, the MRI's intensities are normalized using that histogram as a reference. The skull-stripped MRIs are used and DWI is coregistered to SWAN using ANTS software, producing images of size $512\times512$ in the $(x,y)$-plane and $z$ varies between patients. As input for the model, we take $256\times256\times s$-size crops, where $s$ is the chosen number of slices. To reduce $x$ and $y$ the center of mass is calculated to add around 128 pixels in all dimensions (arriving at a size of $256\times256$ around that center). As DWI and SWAN have different resolutions and indeed the lesion is normally a bigger region than the thrombi, the crops in $z$ are done in the original resolution, taking the corresponding slices. Having $s$ slices from SWAN and $s$ slices from DWI means that we are seeing a bigger brain region in the diffusion modality, which allows the model to have an increased attention impact.

## 2.2. Methodology: AttLLSTM

To detect the thrombi, the neurologists look for the lesion as the first step. From the location of the stroke consequence (damaged tissue) they know where to look for its cause (only one hemisphere is affected for example). Trying to mimic that reasoning, we propose AttLLSTM which merges DWI (where the lesion is visible) and SWAN and PHASE (where the thrombus is seen) using cross-attention. An improved version of the CLSTM (Shi et al., 2015) that we denote as Logic long-short time memory (LLSTM) is used to segment the thrombi. The longitudinal direction matches the time using $s$ slices (the prediction is done by calculating the cell and hidden states with the previous slices). Finally, the segmentation is improved by merging the full lesion and thrombi predictions using a post-processing module. The details of AttLLSTM are shown in Fig. 1.

---

1. It measures the extent of early ischemic changes in anterior circulation hyperacute ischemic stroke. A point is subtracted if it touches one of the 10 brain divisions

**Figure 1:** AttLLSTM. The thrombus is segmented using a cross-attention module between DWI, SWAN and PHASE followed by the LLSTM architecture.

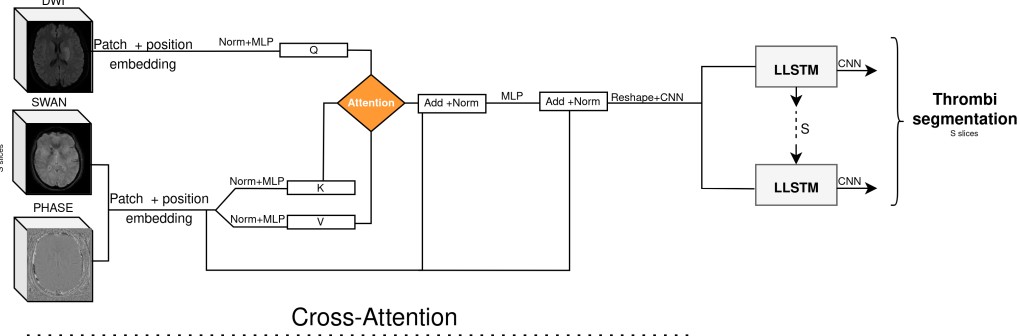

### 2.2.1. Cross-attention

The cross-attention module merges the diffusion modality DWI (as it is just to guide where the affected area is, only one diffusion modality is used) and the susceptibility ones (where the thrombi information is). It is computed as in CrossVIT (Chen et al., 2021). First, we apply a patch embedding ($p_1$ (DWI), $p_2$ (SWAN, PHASE)) of the images. We compute $Q$ from the embedding of the diffusion modality and $K$ and $V$ from the embedding of the concatenation of susceptibility ones, normalizing and applying an MLP. The scaled dot-product attention (Vaswani et al., 2023) is then calculated from $Q$, $K$, and $V$. As $p_1$ and $p_2$ are different values in our case, the smaller embedding output (larger patch size) is resampled. The usual Transformer Encoder module is used where a residual connection is done with a normalization layer, followed by an MLP and another residual connection and normalization. After this module, we reshape the output (going back to a 3D image) and a convolution layer (CNN) of size 7×7 is included (having the same output channels as input ones) followed by Elu activation. All these operations are applied per slice (in 2D).

### 2.2.2. Logic LSTM

Denoting $V||^l S$ as the concatenation along the axis $l$ of the two tensors $V$ and $S$, as $\circ$ the Hadamard product and as $\mathcal{L}(A) = W * A + b$ the classic convolution with a kernel $W$ of size $w \times w$, CLSTM (Shi et al., 2015) is defined by the following equations:

$$A = c_t||^4 h_t||^4 x_t \quad (1) \qquad o_t = \sigma(\mathcal{L}_o(A)) \qquad (4) \qquad c_{t+1} = f_t \circ c_t + i_t \circ d_t \quad (6)$$
$$f_t = \sigma(\mathcal{L}_f(A)) \quad (2) \qquad d_t = \tanh(\mathcal{L}_d(A)) \quad (5) \qquad h_{t+1} = o_t \circ \tanh(c_{t+1}) \quad (7)$$
$$i_t = \sigma(\mathcal{L}_i(A)) \quad (3)$$

where $x_t$ is the input image of size $n_1 \times n_2 \times 1 \times n_{4_{\text{Input}}}$. The cell state $c_t$, which is the model's memory, and the hidden state $h_t$, which is the model's output, are of size $n_1 \times n_2 \times 1 \times n_{4_{\text{hidden}}}$. The forget gate $f_t$ and input gate $i_t$ decide what is erased from and written to the memory $c_t$. The output gate $o_t$ determines which parts of the memory are retrieved to create the output. At time $t$, the model has access to its memory of all previous steps, in our case, the longitudinal direction matches time. To reduce the number

**Figure 2:** LLSTM. The convolution operation is replaced by the Logic one. This operation reduces the trainable parameters as it is a concatenation of two parts ($a_1$, $a_2$) where smaller convolutions are applied and a pooling layer is included.

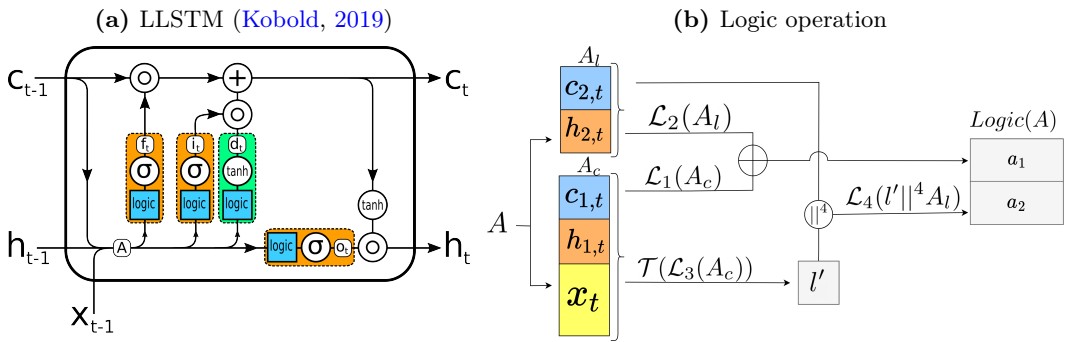

**(a)** LLSTM (Kobold, 2019)  **(b)** Logic operation

of parameters and increase the receptive field we propose the operation $Logic(A)$, which replaces $\mathcal{L}(A)$, denoting the architecture as LLSTM (Kobold, 2019) as shown in Figure 2.

The operation is reduced to the concatenation of a convolution part ($a_1$) and a logic part ($a_2$). The convolution operation is only applied on a part of $A$ ($A_c$), reducing considerably the number of learned parameters. To do so, we first slice $h_t$ and $c_t$ in $h_{1,t}, h_{2,t}$ and $c_{1,t}, c_{2,t}$.

$$c_t = c_{1,t}||^4 c_{2,t} \qquad h_t = h_{1,t}||^4 h_{2,t}, \tag{8}$$

i.e. the hidden state $h_t$ is split into a convolution part $h_{1,t}$ with $n_c$ channels and a logic part with $n_l$ channels and likewise for the cell state $c_t$. Using them we define the splits of $A$, $A_c$ and $A_l$ as following

$$A_c = c_{1,t}||^4 h_{1,t}||^4 x_t \qquad A_l = c_{2,t}||^4 h_{2,t}. \tag{9}$$

Considering that $\mathcal{L}_i$ are convolution layers with weights $W_i$ and $b_i$ and in particular $\mathcal{L}_2$ has $b_2 = 0$, we obtain the convolution ($a_1$) and the logic ($a_l$) part and $Logic(A)$ as following:

$$a_1 = \mathcal{L}_1(A_c) + \mathcal{L}_2(A_l) \qquad a_2 = \mathcal{L}_4(\mathcal{T}(\mathcal{L}_3(A_c))||^4 A_l) \tag{10}$$

$$Logic(A) = a_1||^4 a_2 \tag{11}$$

where $\mathcal{T}$ is put for the Transfer Layer, which produces the same output size as the input size. $\mathcal{L}_1, \mathcal{L}_3$ and $\mathcal{L}_2, \mathcal{L}_4$ are $w \times w$ and $1 \times 1$ convolutions respectively. Finally, denoting $(V_{i_1 i_2 i_3 i_4})_{i_k \in [r_1 \ldots r_2]}$ as the slicing $V$ by setting the dimension $i_k$ to a specific range $r_1, \ldots, r_2$ or a single value $r_1$, $\mathcal{T}$ is defined as follows:

$$(\mathcal{T}(I)_{i_1 i_2 i_3 i_4})_{i_4=u} = \mathrm{maxpool}\left((I_{i_1 i_2 i_3 i_4})_{i_4=u}, p\right), \text{where } u = im + j \tag{12}$$

with $i \in \{1, \ldots, k\}, j \in \{1, \ldots, m\}$. $I$ is of dimensions $n_1$, $n_2$, $n_3 = 1$, $n_4 = n_{4_l} = km$ where the multiplicity $m$ is a positive integer. We choose $p$ as powers of 2 using $p = \frac{2n_1}{2^i}$. For example, having $n_4 = 6$ and $m=3$, we have $k= 2$ so we apply a pooling layer on the first three features, and another one in the last three ones, each one with different sizes. So, this layer performs a max pooling (Ranzato et al., 2007) operation on each channel separately, with varying window sizes ($p$).

The convolution result ($a_1$) can be seen as the original CLSTM, being applied to part of $A$ ($\mathcal{L}_1(A_c)$) and receiving information from the logical part ($\mathcal{L}_2(A_l)$) as additional bias. The $\mathcal{T}$ function moves features across the image in a series of different neighborhood sizes, from local to global. This allows the receptive field of the gates to be as large as the whole input image (which allows doing the concatenation with $A_l$) and captures the spatial distance as well. In addition, a double pass is done to not start with zero memory and to replicate a bidirectional recurrence method but without adding parameters. The first pass is done to save all the memory states, so the prediction is done using them in the second one (Kobold, 2019). Finally before the prediction, a convolutional layer is included of size $1\times1$ with output channels 2 (number of classes).

### 2.2.3. Lesion segmentation and post-processing module

The lesion and thrombi segmentation are merged to reduce the possible false positive thrombi detections. For this purpose, nnUnet (Isensee et al., 2021) is trained using ADC and DWI as inputs (the modalities in which the hyper-acute lesion is seen (Tsai et al., 2014)), reaching a Dice of 0.72 in average and roughly a 100% percent detection rate. More details about these results are in Appendix A.

Two steps are used as postprocessing. First, we choose the object that is closer to the lesion prediction (if there is one), reducing false positives. Using objects with a certain size (the ones smaller than 20 pixels are eliminated), the Euclidean distance between the center of mass of both segmentations is computed. In order to relax constraints, we select the closest object to the lesion with at least $N$ pixels of distance from the others. This produces as output in most cases just one object. Finally, the pixels that are around the predicted one are included reducing the pixel classification threshold ($T$), increasing the size of the predicted thrombus.

## 2.3. Training process and metrics

The training configuration is described in Table 2. We use flipping (in three dimensions) and Gaussian noise (each one is chosen in every iteration with 40% of probability) as data augmentation. As the thrombus is a small object, we use per iteration a crop including the thrombi and another without to manage the unbalanced dataset. We have a sequence of slices containing the full thrombus (successively as the thrombi is a dense object), some other slices just before, and after the thrombus (all of them without thrombi). So per iteration, one crop of $s$ slices is chosen from the first group of crops and another from the other two, both for the same patient.

**Table 2:** Training process. Adam optimizer is used with learning rate (lr) 0.01

| Batch size | Batch's crops | $p_1$ | $p_2$ | Att embed | $n_c$ | $n_l$ | $m$ | $n_4$ | Train, Val, Test (%) | Loss function | $T$ | $N$ |
|---|---|---|---|---|---|---|---|---|---|---|---|---|
| 2 | 4 | 4 | 32 | 32 | 4 | 8 | 3 | 32 | 70,20,10 | Cross Entropy | 0.3 | 20 |

The metrics used are the Dice score, which is calculated pixel-wise and measures the overlap between the prediction and the ground truth, the average count and size of false positives (FP) and false negatives (FN) (in pixel count), and the detection rate (det.) calculated by patients, being one if at least one pixel's prediction overlaps the ground truth.

## 3. Results

As the number of slices is a critical hyper-parameter on the recurrent methods presented (in nnUnet the software chooses the crops by itself), a specific study on that parameter is done. The results using several slices are in Table 3 for proximal (P) and distal (D) occlusions.

**Table 3:** Results using different numbers of slices for segmenting the thrombi using AttLLSTM with CHSF and MATAR dataset.

| Slices | False Positives | | False Negatives | | Dice | Det. |
|---|---|---|---|---|---|---|
| | Count | Size | Count | Size | | |
| | P \| D | P \| D | P \| D | P \| D | P \| D | P \| D |
| 10 | 1.3\|0.9 | 74.1\|102.7 | 0.3 \|0.3 | 292.7\| 79.4 | 0.48\|0.51 | 0.9\|0.91 |
| **12** | **1.0\|0.6** | **49.8\|63.3** | **0.2\|0.3** | **0.2 \|79.4** | **0.55\|0.54** | **1.0\|0.91** |
| 14 | 3.3\|2.3 | 79.4\| 82.8 | 0.2 \|0.4 | 0.2 \|108.5 | 0.59\|0.42 | 1.0\|0.85 |

Using $s$ equal to 12 for all the recurrent methods, as it produces the best result using AttLLSTM, we compare the results with some of the SOTA methods for the thrombi segmentation in Table 4. nnUnet is trained using the general procedure and all our contributions are tested: first, just LLSTM and CLSTM (adding at the beginning a CNN with 5×5 filters instead of the attention), the attention module, and the two post-processing techniques (lesion and threshold (Thr)). As the other methods proposed for the thrombi use the hemisphere position (Tolhuisen et al., 2020) or the reference path through the brainstem as annotations (Zoetmulder et al., 2022), we do not include them.

**Table 4:** Results using different architectures and configurations for segmenting the thrombi using 12 slices for the recurrent methods. The best model is highlighted in bold and the best model without post-processing techniques is underlined.

| Model | Datasets | Lesion | Thr | False Positives | | False Negatives | | Dice | Det. |
|---|---|---|---|---|---|---|---|---|---|
| | | | | Count | Size | Count | Size | | |
| | | | | P \| D | P \| D | P \| D | P \| D | P \| D | P \| D |
| nnUnet | CHSF+MATAR | | | 0.6\|1.1 | 37.1 \|222.4 | 0.4\|0.5 | 373.6\|125.3 | 0.46\|0.45 | 0.7\| 0.7 |
| CLSTM | CHSF+MATAR | | | 3.8\|2.6 | 178.1\|153.5 | 0.2\|0.4 | 0.2 \| 28.1 | 0.39\|0.33 | 1.0\|0.81 |
| LLSTM | CHSF+MATAR | | | 0.8\|1.7 | 111.9\| 34.9 | 0.2\|0.6 | 0.2 \|136.4 | 0.48\|0.36 | 1.0\|0.64 |
| AttCLSTM | CHSF+MATAR | | | 1.2\|1.6 | 43.81\| 99.3 | 0.3\|0.4 | 141.1\|100.7 | 0.41\|0.38 | 0.9\|0.72 |
| AttLLSTM | CHSF | | | 3.8\|3.0 | 111.9\|101.5 | 0.2\|0.6 | 0.2 \|136.4 | 0.50\|0.27 | 1.0\|0.64 |
| AttLLSTM | CHSF+MATAR | | | 1.0\|0.6 | 49.8 \| 63.3 | 0.2\|0.3 | 0.2 \| 79.4 | 0.55\|0.54 | 1.0\|0.91 |
| nnUnet | CHSF+MATAR | ✓ | | 0.0\|0.7 | 0.0 \|161.3 | 0.4\|0.5 | 373.6\|130.9 | 0.48\|0.51 | 0.7\| 0.7 |
| AttCLSTM | CHSF+MATAR | ✓ | | 0.5\|0.9 | 19.4 \|109.8 | 0.3\|0.6 | 141.1\|134.4 | 0.43\|0.36 | 0.9\| 0.6 |
| AttLLSTM | CHSF+MATAR | ✓ | | 0.0\|0.1 | 0.0 \| 8.5 | 0.2\|0.3 | 0.2 \| 79.4 | 0.52\|0.58 | 1.0\|0.91 |
| nnUnet | CHSF+MATAR | ✓ | ✓ | 0.0\|0.8 | 0.0 \|177.9 | 0.5\|0.5 | 373.6\|130.9 | 0.49\|0.51 | 0.7\| 0.7 |
| AttCLSTM | CHSF+MATAR | ✓ | ✓ | 0.5\|0.9 | 59.9 \|245.8 | 0.3\|0.6 | 141.1\|134.4 | 0.54\|0.32 | 0.9\| 0.6 |
| **AttLLSTM** | CHSF+MATAR | ✓ | ✓ | **0.0\|0.1** | **0.0 \| 8.5** | **0.2\|0.3** | **0.2 \|79.4** | **0.63\|0.59** | **1.0\|0.91** |

nnUnet struggles to segment the thrombi: less than 80% of the patients are detected. LLSTM outperforms CLSTM in the Dice score, even though the detection rate is lower. Adding to both recurrent methods the attention module increases the performance. AttLLSTM has the highest detection rate (having 91% in the smaller cases) and Dice. Without using the distal occlusions, AttLLSTM misses 40% of the patients and including them in the training improves all metrics. For the post-processing techniques, adding the lesion (Lesion) reduces the number of false positives in more than half of them but also reduces the detection rate for distal cases (some wrong close objects are chosen) for AttCLSTM.

**Figure 3:** Visual prediction examples. Original slice and zoom versions are shown. The ground truth (Gt) and the predictions (pred) are shown in the zoomed version. The first row is concerned with proximal thrombus, and the second one with distal thrombus.

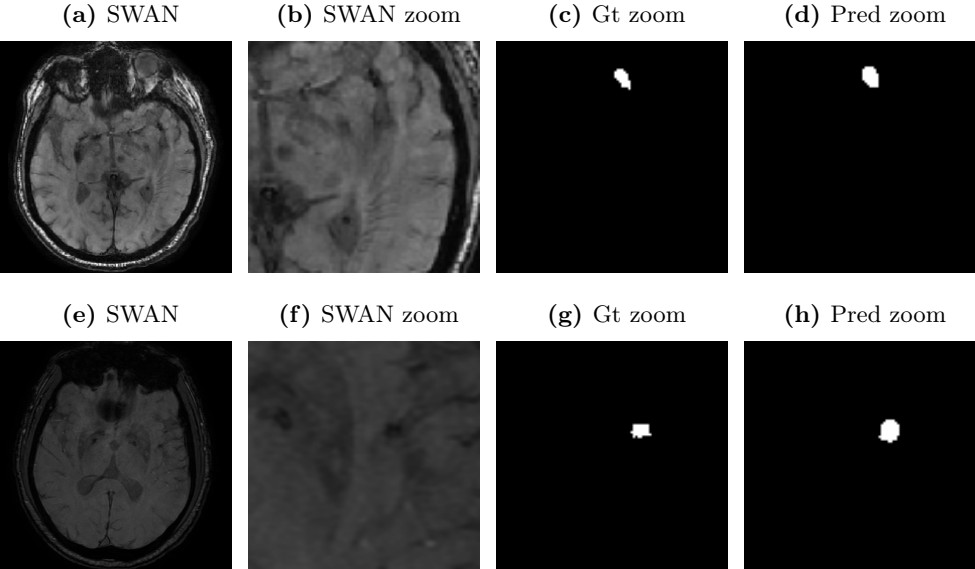

| **(a)** SWAN | **(b)** SWAN zoom | **(c)** Gt zoom | **(d)** Pred zoom |

| **(e)** SWAN | **(f)** SWAN zoom | **(g)** Gt zoom | **(h)** Pred zoom |

Finally, the threshold technique (Thr) allows to have on average a Dice of 0.43 or higher for all. AttLLSTM gets the best performance (0.61 as Dice) detecting almost all the patients (missing less than 10% of distal thrombi). Indeed, the model is already better without the post-processing comparing it with nnUnet/AttCLSTM +lesion+thr in all metrics. To test the robustness of the model, the control MRI segmentation results are present in Appendix B. Figure 3 presents some predictions obtained, the original SWAN, zoomed, and predictions are shown, for a proximal thrombus (a,b,c,d) and distal one (e,f,g,h).

## 4. Conclusion

We have proposed a model relating the lesion and the thrombi segmentation on MRIs that outperforms the other methods tested. A cross-attention between the modalities as a recurrent model where the longitudinal direction matches time and a way of merging the lesion and the thrombus is defined. AttLLSTM detects almost all thrombi present in stroke patients in less than 3 minutes, producing promising results in the hyper-acute phase, for distal and proximal clots. This detection could save time, help to choose a treatment or estimate the survival of the patients.

But, due to the lack of public datasets, the multicentric robustness cannot be tested, even though augmentations and normalizations have been defined to overcome that problem. MRIs from different vendors will not only produce different images due to calibration but the susceptibility modalities can also be different: SWI instead of SWAN. The two postprocessing proposed techniques could be done differently: for example, merging the lesion and thrombus with a learning method or improving the Dice with another technique.

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

## Appendix A. Lesion segmentation

In Table 5 we can see the results for the lesion. nnUnet is used as a reference model, using exactly the training and validation procedure available in the library (just including our splits). It outperforms LLSTM using just the proximal dataset (CHSF). So nnUnet is trained on both datasets, arriving at a Dice of 0.72 on average and close to a 100% percent detection rate. Notice that there is a considerable difference between the performances on the proximal (P) and distal (D) datasets. That model is used for the post-processing module in AttLLSTM.

**Table 5:** Test set results using LLSTM and nnUnet for lesion detection on both datasets.

| Model | Datasets | False Positives | | False Negatives | | Dice | Det. |
|-------|----------|-------|------|-------|------|------|------|
| | | Count | Size | Count | Size | | |
| | | P \| D | P \| D | P \| D | P \| D | P \| D | P \| D |
| LLSTM | CHSF | 0.8\| | 82.8\| | 1.4\| | 47.5\| | 0.70\| | 1.00\| |
| nnUnet | CHSF | 3.7\| | 33.2\| | 1.9\| | 5.64\| | 0.75\| | 1.00\| |
| **nnUnet** | **Both** | **1.1\|0.2** | **19.3\|9.7** | **1.3\|2.4** | **23.9\|14.6** | **0.77\|0.67** | **1.00\|0.95** |

## Appendix B. AttLLSTM robustness using control MRIs

The annotations for the control MRIs are available for MATAR dataset. This MRI is taken just after the treatment is given, to evaluate the effect of it. In some cases, the thrombus completely disappears, and in other cases, it can remain almost in the same position so another treatment is used. To formalize the different cases we can define the distance between the initial thrombus (thrombus$_1$) and the one after the treatment (thrombus$_2$):

$$D = \text{dist}(\text{thrombus}_1, \text{thrombus}_2) \tag{13}$$

Depending on its value, we have a different treatment outcome:

- If $D = 0$: No treatment effect

- If $D = \infty$: The treatment makes the thrombus to disappear.

We evaluate our model in the most similar cases to the first MRI (where the distance is smaller) and the cases where the treatment effect is higher (the distance is bigger). We use the distance quartiles $q_{25}$ and $q_{75}$ to obtain these patients, only for the patients where the distance can be calculated (the thrombus is present in both MRIs). In Table 6 we can see the results.

Our model maintains a comparable performance when the treatment has less effect (smaller distance), detecting more than 80% of the patients but it just detects the 25% when the treatment produced a bigger effect (having a Dice of 0.51 and 0.1 respectively). The thrombus movement changes the relationship between the modalities; for instance, the relationship between the lesion and the thrombus is not the same anymore: the treatment can dissolve the thrombus but the lesion remains the same, as it is the damaged tissue.

**Table 6:** Results using DWI, SWAN, and PHASE to detect thrombus using AttLLSTM. For the control MRIs a total of 80 patients are used (the thrombus is still present) and 30 patients satisfy the distance conditions. All the results are obtained with no post-processing.

| Model | Evaluation (MATAR dataset) | False Positives | | False Negatives | | Dice | Det. |
| --- | --- | --- | --- | --- | --- | --- | --- |
| | | Count | Size | Count | Size | | |
| AttLLSTM | 1st MRI | 0.6 | 63.3 | 0.3 | 79.4 | 0.55 | 0.91 |
| AttLLSTM | Control MRI ($D < q_{25}$) | 0.6 | 19.3 | 0.4 | 78.8 | 0.51 | 0.81 |
| AttLLSTM | Control MRI ($D > q_{75}$) | 1.1 | 65.1 | 1.0 | 175.1 | 0.10 | 0.25 |

Because of that, the post-processing techniques are not included in this evaluation. We can conclude that our model maintains a similar performance segmenting the thrombus when the treatment effect is low.

