# OpenReview forum: "A recurrent network for segmenting the thrombus on brain MRI in patients with hyper-acute ischemic stroke"
_MIDL.io/2024/Conference — MIDL 2024 Poster_

### Official Review · Reviewer_cheR · 2024-02-28

**Confidence:** 5
**Preliminary Rating:** 2
**Final Rating:** 3.5

**Summary:**

The authors modified Convolutional Long-short-term Memory (CLSTM) and built Logic long-short time memory (LLSTM) as recurrent network for segmenting thrombus on brain MRI in patients with hyper-acute ischemic stroke. This paper utilizes two datasets: MATAR (distal) and CHSF (proximal).

**Strengths:**

1. The manuscript is easy to follow;
2. The idea of introducing cross-attention is interesting;
3. The proposed framework utilizes different modalities within Brain MRI;
4. Authors clearly explained their data preprocessing steps;

**Weaknesses:**

1. In section 2.1, the CHSF dataset is explained in detail, whereas the description for the MATAR dataset is limited to Table 1;
2. Combining both datasets, they only have 188 data points. Even after splitting the data into training, validation, and testing sets, the dataset remains very small;
3. If the dataset is too small, it is hard to understand the robustness of the proposed framework; The authors could evaluate their framework on more target datasets;
4. Table 2 displays hyperparameters, but a more thorough ablation study is necessary. For instance, is selecting only 12 slices sufficient? Furthermore, how were these 12 slices chosen?
5. In Table 3 and 4, does "Both" mean utilizing MATAR and CHSF datasets together?
6. The performance for the proposed AttLLSTM is not showed for lesion detection.
7. Did the author consider utilizing vision transformers instead of LSTM? Can vision transformer perform better than LSTM here?
8. The authors mentioned couple of methods in their related study. I would suggest authors to compare their proposed framework with other state-of-the-art methods.

**Detailed Comments:**

Please check the points mentioned in the section "Weaknesses".

**Justification Of Final Rating:**

The authors have addressed most of my questions, and they have also made some adjustments in their Appendix. However, the dataset's size remains a limitation, making it challenging to draw definitive conclusions.

**Justification Of The Preliminary Rating:**

Evaluating the robustness and generalization of the proposed framework is challenging due to the limited dataset used by the authors for their evaluation. While the performance appears to be satisfactory, it does not provide a comprehensive understanding. Including more diverse target tasks in their experiments would provide a clearer and more comprehensive picture of the framework's capabilities.

**Questions To Address In The Rebuttal:**

Please check the points mentioned in the section "Weaknesses".

---

> ### Author Response · Authors · 2024-03-17
> **Rebuttal to reviewer 3**
>
> Dear Reviewer
>
> Thank you very much for all the remarks. Concerning the  weaknesses of our text:
>
> - All section 2.1 talks about MATAR and CHSF, I think the misunderstanding was because the hospital (Centre Hospitalier Sud Francilien) acronym and the name of the first dataset "CHSF" were the same. Now it is clarified in section 2.1.  More information about the dataset (description of geometries, modalities, the groundtruths etc) is provided.
>
> - It is true that we just disposed of 188 patients but several experiments were done (all possible configurations are now available either in Appendix B or Table 4. Also, the evaluation on control MRIs are included now in Appendix C (in 30 more patients).
>
> - As the proposal is specific for thrombus detection, as the biological relationship between the lesion and the thrombi are key points of the model and prediction, we cannot use another target dataset. There is no public dataset for this purpose but as mentioned before now the evaluation on control MRIs is added including additional configurations and also a SOTA model to assess the model's robustness.
>
> - Indeed, the number of slices is a critical hyperparameter, we have added. Table 3 presents the experiments done to optimize its choice (section 3).
>
> - Yes, "both" means MATAR and CHSF. Both names appear explicitly now in the tables  to clarify this.
>
>
> - AttLLSTM is only for thrombus segmentation. It uses a set of MR sequences and it cannot be used for lesion segmentation as the idea of this architecture is to guide the thrombi prediction (which is more difficult due to its size) by using the lesion (bigger and easier to segment) imitating the neurologist procedure. Knowing where the lesion is they look for the thrombus. As the lesion section was confusing and not many experiments were done, as it is just used as a support, the results are now just mentioned in the methodology (section 2.2.3) and all the details are in Appendix A
>
> - About vision transformers, due to the lack of data and the immense size of the architecture, we decided to not test it. Our proposal tries to imitate a part of its mechanism (the attention module) but with a lighter architecture, to reduce the overfitting risk.
>
> - Some of the state-of-the-art methods use other information as input (brain hemisphere or brainstem) that we do not have and in addition, we wanted to do a model free of these annotations (as they are timely and expensive). But the comparison with Multidirectional Unet is now included in Table 4 as well as all the possible configurations.

---

### Official Review · Reviewer_YCba · 2024-02-28

**Confidence:** 3
**Preliminary Rating:** 2
**Final Rating:** 3.5

**Summary:**

The paper addresses the problem of segmenting the thrombus within MRI images of patients suffering from acute ischemic stroke. Multimodal MRI are used as input (ADC, DWI, SWAN and PHASE). A lighter version of Convolution LSTM is proposed through the definition of a Logic operation. The induced Logic LSTM (LLSTM) model is used to segment the lesion from ADC and DWI images on one hand, and the thrombi from DWI, SWAN and PHASE images on the other hand, after these modalities have been combined through a cross attention module. Three post-processing steps are further applied: proximity of the thrombus to the lesion, and hysteresis thresholding, together with size filtering. The method is tested on two datasets including both proximal and distal thrombi (resp. 63 and 125 patients) and is shown to outperform both nnUnet and CLSTM, with very few detection errors.

**Strengths:**

The adressed clinical question is of high importance, challenging and few solutions have been proposed. Even though the description of the proposed Logic LSTM in section 2.2.2 is quite technical, and its readability could certainly improved, the authors made a commendable effort to explain the reasons behind the proposed changes, and their impact. The study in section 3.2 is very helpful in assessing the relative importance of proposed contributions.

**Weaknesses:**

The paper is very difficult to read, and its writing should be dramatically improved. Overall some critical information is missing, and many other pieces of information appear to be scattered in the text. A careful re-reading might help reorganize these to improve the text. The main points are:
- MRI modalities are referred to by acronyms only. The authors should be aware that their paper might be of interest to researchers looking for methods to segment small objects from multiple modalities, which is a broader audience than neuroradiologists interested in thrombi delineation to treat AIS. Furthermore, information about their complementarity should be presented at the beginning of the paper (instead of the first sentence of section 2.2.1 for example). Also, why just use DWI for cross attention and not embed DWI and ADC together (the first sentence of 2.2.1 is about diffusion modalities (plural))?
- the databases are partially described but this should be improved. The reader should not have to read a clinical publication to understand what MATAR is, or the conditions to access the data. DWI and SWAN image geometries are provided but what about ADC and PHASE? Are there healthy patients in the cohorts?
- Section 1.2 provides a rather complete overview of the previous methods for lesion segmentation, but it is mainly descriptive. What are the limitations of these methods? Why discard these methods and only compare to nnUnet in section 3.1? The separation between lesion and thrombus segmentation is unclear.
- ISLES uses FLAIR and DWI for lesion segmentation. The authors use ADC and DWI. Is it merely practical (modalities present in the databases) or are there more fundamental reasons?
- see detailed comments for more examples.

In the LSTM model, the longitudiinal direction serves as time. What about causality? Is there a difference between top to bottom or bottom to top directions? Would there be an interest in coupling both derived models (in an ensemble way for example)?

A lot of preprocessing is performed (see end of section 2.1). I suppose it is required at inference time. Is it fully automatic? Does it require some manual intervention? In such cases, is it compatible with the clinical emergency setting? Cropping operations are also somewhat unclear. For example in section 2.3 the authors refer to "the crop that involves all the thrombi". Is it always possible with 12 slices? Is it always unique? How are these crop operations involved at inference time?

In section 3.1, nnUnet is better than LLSTM to segment the lesion. But in section 3.2, AttLLSTM is better for thrombus segmentation, in particular if both proposed post-processing steps are applied. These steps seem to be applicable to the result of nnUnet, exploiting the segmentation of the lesion by nnUnet. Unless I am mistaken, this configuration (nnUnet + Lesion + Thr) should be included in the Table 4. Also both the attention module and post-processing steps are compatible with CLSTM. The induced configurations should be included in Table 4.

- Reproducibility (optional): it would be very helpful to better grasp the details of the proposed model if a code would be available to the community

**Detailed Comments:**

- on page 2: General Electrics should be replaced by GE Healthcare (GE has officially replaced General Electric 25+ years ago)
- CLSTM is mentioned on page 3, but the reference is only given on page 4.
- Figure 1 is a schematic of the proposed method, but there is no mention of LLSTM. A line connecting SWAN to the cross-attention module is missing
- At the beginning of section 2.2 (p. 3), the authors write that "space is used as time steps", and later (p. 5) that "the time t becomes the slices along the $x_3$ axis. It would be simpler in my opinion, to write in the first place that the longitudinal direction matches time in the LSTM architecture.
- at the beginning of section 3.1, the authors assume that nnUnet outperforms LLSTM "probably due to all the preprocessing and post-processing incorporated in nnUnet". This claim is not supported by any evidence and should be avoided. If such pre/post-processing steps are vital to the quality of the results, why did not the authors include them?
- The units for the size metric should be given (mm3 or voxel count?)
- the acronym AttLLSTM appears in section 3.2 as "our proposal". It should be introduced when the proposed method is actually described.
- some mishaps in the English writing (e.g. Section 2.2.2, first sentence "Denoting as ... the concatenation" -> "Denoting ... as the concatenation", or Section 3.2, first sentence: "... nnUnet is used as before and CLSTM" seems to be missing some words)

**Justification Of Final Rating:**

I'd like to thank and commend the authors for the work they have put in the revised paper. In my opinion, the paper is much easier to read and apprehend. However I still have some concerns about the significance of the reported results and the actual impact of the attention to lesion in the proposed model, which exists indeed, but still requires to be boosted by post-processing.

**Justification Of The Preliminary Rating:**

I mostly consider the readability of the paper should be improved. I ponder whether the lesion detection experiment is useful, especially since the proposed solution is not compared to any prior work on lesion segmentation. That would make some space for such improvements and supplementary experiments.

**Questions To Address In The Rebuttal:**

- propositions to improve the readability of the paper (see weaknesses above)
- discuss whether section 3.1 is actually useful. If not, expand section 3.2 with results including nnUnet + Lesion + Thr, and a version of AttCLSTM (+ Lesion + Thr). Unless I am wrong and the latter combination is impossible.

---

> ### Author Response · Authors · 2024-03-17
> **Rebuttal to reviewer 2**
>
> Dear Reviewer,
>
> First of all, thank you for all the remarks and details mentioned in the review. We have made several improvements and the detailed comments are mentioned below:
>
> - MRI modalities:  all acronyms are included now, the explanation of why each modality is used and their complementarity in section 2.1 and why only DWI is used for the attention module in section 2.2.1
>
> - Database description: section 2.1 describes both datasets (MATAR and CHSF), I think it was not clear to include the same acronym for the hospital (CHSF) and the dataset CHSF. All MRIs come from Centre Hospitalier Sud Francilien with similar conditions. This clarification is now included as well as the modality geometries. All the patients had a stroke but the control MRIs are available only for the distal ones, the results on these MRIs are now included in the appendix.
>
> - Section 1.2: the limitations of the lesion segmentation methods as well as a clear separation between lesion and thrombus are precise in the new paper version. Not all the lesion methods are tested as the main contribution of the paper is the thrombus segmentation, in this new version of the paper the main objective is clear.
>
> - Why ADC and DWI? The lesion delineation is done based on ADC and DWI in the hyper-acute phase. When there are old lesions they are visible in FLAIR, probably that's why it is also included in some studies (when there are strokes after several months), but it is not our objective as we are focusing on the actual brain lesions segmentations. This clarification is now included in section 2.1 and when the lesion segmentation results are shown in section 2.2.3
>
> -Causality: the model was trained by doing a double pass to imitate the bidirectional LLSTM without adding parameters to train: with the first one the memory is calculated to be used for the prediction using the second pass (it is mentioned at the end of the section 2.2.2)
>
> - Preprocessing: the preprocessing is fully automatic, the modality registration using ANTS is the longest stage (up to 3 minutes, depending on the patient) but it is compatible with the clinical emergency setting. As the thrombus is a small object, we use per iteration a crop including the thrombi and another without to manage the unbalanced dataset. We have a sequence of slices containing the full thrombus (successively as the thrombi is a dense object), some other slices just before, and after the thrombus (all of them without thrombi). So per iteration, one crop of 12 slices is chosen from the first group of crops and another from the other two, both for the same patient. This clarification is now included in section 2.3 and why 12 slices are chosen in section 3. The number of slices is a hyperparameter chosen as it was the best one obtained after several experiments (table 3 in section 3). Not always with 12 slices is all the thrombi is inside, so each time a crop center is chosen randomly and used.
>
> - More configurations: AttCLSTM results are now available in table 4 and for the post-processing methods, as the methods can only improve the Dice and reduce the false positives, we show in table 4 only the results for the best model (especially having a higher detection rate) but all the configurations and results are now included in appendix B (page 12), as it is possible to include them.
>
>
> - General electrics is now replaced by GE Healthcare
>
> - CLSTM is correctly referenced now
>
> - Figure 1 is improved giving all the details now
>
> - The sentence "the time t becomes the slices along the axis" is replaced by "longitudinal direction matches time" and included at the beginning of 2.2 and illustrated in Figure 1
>
> - Not many experiments were indeed done for the lesion as it is only used as support for the thrombus detection, so the section for the lesion results is canceled and the results are mentioned in section 2.2.3 (with all the details  in Appendix A)
>
> - The units are now clarified in section 2.3 (metrics)
>
> - AttLLSTM is now mentioned from the beginning in the methodology section (section 2)
>
> - The mishaps are solved now
>
> About the questions to address
>
> - The readability of the paper has improved with all the changes done and thanks to the figures added
>
> - Section 3.1 is now in the appendix but the model used is mentioned in the methodology. The experiments of why 12 slices are chosen and the results using AttCLSTM are now in section 3. All the configurations with the post-processing (nnunet+lesion+thr, attclstm +lesion+thr) are available in the appendix

---

> > ### Comment · Reviewer_YCba · 2024-03-27
> >
> > I want to thank the authors for the extensive work that has been done to take my comments into account and improve their paper. I feel that it is much clearer and easy to read now. The rationale and the imaging modalities are now clear, Figure 1 is way more informative, so is the addition of Figure 2.
> >
> > However, I conquer with one of my fellow reviewers that the size of the database does not allow for any definite answer, even though a positive trend appears in the reported results. Furthermore, the impact of the attention module is difficult to circumscribe: it is supposed to leverage the lesion location to help locating the thrombus, but still, selecting the closest detection to the lesion is a necessary post-processing step. This somewhat mitigates the impact of the attention module in my opinion.
> >
> > I have other editing suggestions to make:
> > - p. 2, section 1.1, l1: As the lesion is present brighter -> As the lesion is brighter
> > - use "larger" instead of "bigger" to describe patches
> > - in Table 1, I do not reckon that pixel count is an appropriate metric when images of different resolutions are used. I think pixels are taken in SWAN (PHASE) images for the thrombus, but the lesion is segmented from DWI+ADC which have a different resolution. Using pixel count for both objects does not seem very appropriate to me.
> > - Also, in the same table, as well as in the text, I would reference CHSF first, and then MATAR. This would help the reader make the connection with P/D used later if the same order is kept consistent (CHSF corresponds to P(roximal thrombi) and MATAR corresponds to D(istal thrombi). The same idea could be kept later, e.g. in table 4 where CHSF+MATAR could be used instead of MATAR+CHSF, to be consistent with P+D.
> > - In section 2.2, on page 3, the following sentence is hard to read: "An improved version of the CLSTM... with the previous slices). Please consider splitting it in two: a first sentence to introduce LLSTM, used to segment thrombi; and a second to specify that time is replaced by (matched with) the longitudinal direction, where s slices are considered.
> > - First two columns in table 3 could be removed and the AttLLSTM/MATAR+CHSF setup could be mentioned in the caption.
> > - Adding MD-UNet as another reference method is good, but I cannot find any interest in presenting results only on the CHSF database. The only result on the CHSF database of interest is that of AttLLSTM + Lesion + Thr, which is not reported. But AttLLSTM alone, as done by the authors, enables to highlight the interest of the merged database. In Table 4, I feel the MD-Unet line could be removed, and nnUnet+Lesion+Thr from the appendix should be added, together with AttCLSTM+Lesion+Thr if possible.

---

> ### Author Response · Authors · 2024-03-28
>
> Dear Reviewer,
>
> Thank you again for your remarks. Even though our database is considerably small in terms of the number of patients, we tested all the possible configurations and we include the results for the control MRIs in the appendix, to show how robust is our proposal. The inclusion of the lesion distance is needed as we do not have zero positives and this is key for some next studies (thrombus length, etiology, cause, etc). With some few false positives, these studies could not be done. But in any case, the attention module helps in all the cases (CLSTM and LLSTM) even more than the lesion distance (for AttCLSTM, the lesion distance reduces the detection rate as if there are several false positives, the choice of the closest thrombus to the lesion might be wrong, as the relation between them is not as simple as the closest one).
>
> Concerning the remarks:
>
> - Done
> - Done
> - The sizes are now shown in mm3
> - The order CHSF and MATAR is now consistent in all sections.
> - Done
> - Done
> - Done

---

### Official Review · Reviewer_1SJv · 2024-03-05

**Confidence:** 3
**Preliminary Rating:** 5
**Final Rating:** 5

**Summary:**

In this work, the author introduced a attention based CLSTM model for segmenting the thrombus on the MRI of patients with hyper-acute ischemic stroke. It is significant to note that the inclusion of multiple modalities, such as DWI, SWAN, PHASE are included in a cross-attention manner. It was proposed that with the cross attentioN, CLSTM, post-processing, a over 0.6 DICE score with few False positive can be achieved for this segmentation task.

**Strengths:**

1. It is very reasonable and intuitive to use cross-attention as way to mimic the way neurologists deal with the lesion
2. It provides a detailed logic reasoning of the LSTM deravation and thorough experiments.

**Weaknesses:**

This is an interesting work, however, the organization and figure of the Methodology could be improved. First of all, the paper lacks a overall/detailed figure showing the design of the work. For example, it is not clear for me to fully understand how the modified recurrent network is used in Figure 1. Section 2.2.2 provides detailed logic of LSTM, however, it is unclear to connect it with Figure 1. Secondly, it is interesting to see the author proposes a way to use the slice (s) dimension as the time (t) in recurrent model. However, these related logic (equations in 2.2.2) is better to be related with a visualization.

**Detailed Comments:**

1. For the multi modality problem, the data or modality imbalance (could be extreme imbalance) may be a realistic problem. It will be really interesting and aspiring to see if this model could be improved to deal with the imbalance of the data or modality.
2. Could this model be used for active mining from which hard negative data can be retrieved. it will be good to see the improvement on detecting the hard negatives and summarize the behaviors of hard negative data for better understanding of the stroke scan in MRI.

**Justification Of Final Rating:**

After the revision, my questions are resolved. The author added the required figure showing the design of the work, provided more detailed logic of LSTM used in the brain MRI, and explained the reason of 'slicing' parameters to make the paper more complete. Overall, this work includes the multi-modality and the use LSTM in 3D MRI image segmentation, which is innovative for the community.

**Justification Of The Preliminary Rating:**

Although the visualization in this paper was not very good, the innovation is impressed for me. This work includes the inclusion of using multiple modalities and modified LSTM for the 3D image segmentation, which is a clever way to treat the axial dimension as the traditional time dimension.

**Questions To Address In The Rebuttal:**

1. In the chapter 2.2 Methodology, Figure 1 (Proposed method) is clear for presenting the overview of fusing different modality together. However, it is not clear to relate it to the remaining content of the chapter 2.2. It is better to provide a more detailed figure to show the design of the network model, the post processing module.

2. From the content of 2.3 and Table 2, "A patient’s MRI is divided into three crops: the crop that involves all the thrombi, the one with all the slices before it, and the one after it." It was not clearly stated the reason for the split.

---

> ### Author Response · Authors · 2024-03-17
> **Rebuttal to reviewer 1**
>
> Dear Reviewer,
>
> Thank you very much for the remarks and for the detailed comments, which are possible questions to address for the next research steps. We have made several improvements taking into account the questions to address in the rebuttal mentioned:
>
> - Methodology: we have included an improved overall figure including all the details of the AttLLSTM architecture (page 4). Also, a figure explaining the Logic operation is now included (page 5)
>
> - Crop split: the explanation of the crops was not clear in the previous paper version. As the thrombus is a small object,  to manage the unbalanced dataset, per iteration we use a crop arround the thrombi and another without. Per patient, we have the slices containing all the thrombi (all followed by each other as the thrombi is a dense object), the slices just before that, and the ones after that (all of them without thrombi). So per iteration, one crop of 12 slices is chosen from the first group of crops and another from the other two, both for the same patient. This clarification is now included in section 2.3

---

> > ### Comment · Reviewer_1SJv · 2024-03-24
> >
> > Thank you, my questions have been resolved

---

### Meta-Review · Area_Chair_w7Kc · 2024-04-04

**Recommendation:** Accept (Poster)
**Confidence:** 4

**Metareview:**

The reviewers appreciated the technical novelty, in particular the use of cross-attention, as well as the clininical relevance of the approach. The authors improved the manuscript after rebuttal.

---

### Decision · Program_Chairs · 2024-04-06

Accept (Poster)